# Effects of Coir-Based Growing Medium with Municipal Solid Waste Compost or Biochar on Plant Growth, Mineral Nutrition, and Accumulation of Phytochemicals in Spinach

**DOI:** 10.3390/plants11141893

**Published:** 2022-07-21

**Authors:** Rui M. A. Machado, Isabel Alves-Pereira, Carolina Morais, André Alemão, Rui Ferreira

**Affiliations:** 1Departamento de Fitotecnia, MED—Mediterranean Institute for Agriculture, Environment and Development, Escola de Ciências e Tecnologia, Universidade de Évora, 7002-554 Évora, Portugal; 2Departamento de Química e Bioquímica, MED—Mediterranean Institute for Agriculture, Environment and Development, Escola de Ciências e Tecnologia, Universidade de Évora, 7002-554 Évora, Portugal; iap@uevora.pt; 3Departamento de Química e Bioquímica, Escola de Ciências e Tecnologia, Universidade de Évora, 7002-554 Évora, Portugal; 20carolinam@gmail.com (C.M.); andreacalemao99@gmail.com (A.A.)

**Keywords:** *Spinacia oleraceae*, substrates, alternative organic materials, shoot nutrient content, photosynthetic pigments, phenols, ascorbate, FRAP, DPPH

## Abstract

The use of municipal solid waste compost (MSW) and biochar, two renewable resources with a low carbon footprint as components of substrates, may be an alternative to reducing peat and coir usage. The aim of this study was to assess the suitability of selectively collected MSW and biochar as components of the coir-based substrate to spinach grown. An experiment was carried out to evaluate five substrates, coir and four coir-based blends (coir + biochar + perlite, coir + municipal waste compost + perlite, coir + biochar + pine bark, and coir + biochar + pine bark) with 12% (*v/v*) MSW or biochar and 10% (*v/v*) perlite or pine bark. Spinach seedlings were transplanted into Styrofoam planting boxes filled with the substrate. Each planting box was irrigated daily by drip with a complete nutrient solution. Plants grown with MSW had a higher content of calcium. Shoot Mn increased in the biochar-containing mixes. The shoot dry weight of the plants grown in the different blends was higher than those grown in coir. Fresh yield was higher in mixes with MSW and perlite (3 kg/m^2^) or pine bark (2.87 kg/m^2^). Total phenols and DPPH antioxidant activity were not affected by the substrates. However, shoot ascorbate (AsA) content was higher or equal to those plants grown in coir. MSW and biochar are alternatives to reduce the use of coir and peat.

## 1. Introduction

Nowadays, in soilless culture in the substrate, the main goal is to reduce or replace peat as the constituent of substrates owing to high negative ecological and environmental impacts.

There is increasing pressure on stakeholders (both growing media manufacturers and horticulturists) to significantly reduce their reliance on peat [1]. For these authors, the future of growing media will be based on blends of different components which will be renewable and locally produced. In different studies, Machado [2] and Barcelos [3] have shown that coir can replace peat or peat-based substrate in the production of spinach. Despite being a renewable resource, coir has a significant carbon footprint due to transportation, a high-water requirement for washing coir, and the need to treat wastewater. A strategy to minimize or replace peat and coir could be achieved using selectively collected municipal solid waste compost (MSW) and biochar as substrate components. They are renewable resources produced locally, and the selective collection of municipal solid organic waste is increasing in Portugal. Their use lessens the carbon footprint, keeps organic waste out of landfills, and lessens Portugal’s reliance on importing peat and coir. Organic composts exhibit qualities similar to peat in terms of porosity, aeration, and water-holding capacity [4,5,6], and they have a higher nutrient content. In addition, organic compost has humic acids and fulvic acids that may increase nutrient availability [7].

Biochar could be suitable as a substitute for peat in soilless substrates [8]. Tomato plant heights and bell pepper dry weights increased with the addition of 1.3% and 5% (*w/w*) to a soilless mixture of coconut fiber and tuff (volcanic ash) [9]. However, municipal organic compost and biochar usually present high salinity and/or pH [10,11,12].

The use of blends (coir-based, with municipal solid waste compost (MSW compost) or biochar (75%) by volume in the mixture) boosts the potential of those materials as plant-growing substrate components, but some adjustments are still required to decrease pH and/or electrical conductivity [12]. In addition to those components, pine bark or perlite can be utilized to make the adjustments. Pine bark and the perlite may decrease the electrical conductivity (EC) and pH and enhance the physical properties of the blends. Non-amended pine bark substrate has a low pH (4.1 to 5.1) [13] as well as a low cation exchange capacity (CEC) of 40–75 meq/L [14,15]. However, the cation exchange capacity of the pine bark rose with decreasing particle size [13]. Furthermore, it has a low bulk density [15] and water-holding capacity [14], as well as good stability [16]. Perlite is neutral and it has no buffering capacity and contains no mineral nutrients. Perlite is generally added to the root substrate to increase the proportion of large pores, hence reducing the water-holding capacity and increasing the air-filled pore space. On the other hand, perlite may reduce the substrate’s cation exchange capacity, resulting in a pH drop due to increased proton release [17].

Each mix will have physiochemical properties that can alter the water, nutrients, and air availability in the root zone [18], which can affect yield, mineral and phytochemical accumulation, and antioxidant activity. Plant development [19,20], phytochemical accumulation, and antioxidant activity [20,21] may be affected by substrate type. Therefore, the aim of the present study was to evaluate the effects of coir-based mixes with municipal solid waste compost or biochar and with perlite or pine bark on mineral nutrition, plant growth, and phytochemical accumulation of spinach.

## 2. Results and Discussion

### 2.1. Initial Physical and Chemical Properties of the Media

The average values of pH and EC of the mixes with MSW compost either with perlite or pine bark were higher than the other substrates.

The pH of these mixes was approximately 0.3 points higher than the high end of the adequate range for substrates (6.4 to 6.8) [15,22,23,24]. In the mixes with biochar, the pH average value was similar to the high end of the adequate range (Table 1).

EC average values of the mixes with MSW compost reached high values, from 2.8 and 2.99 d*S* m^−1^ (Table 1), but within the adequate range for substrates (0.75 to 3.5 d*S* m^−1^) [15].

The bulk density average values of the mixes (0.17 to 0.19 g·cm^−3^) were significantly higher in mixes than that of coir (0.11 g·cm^−3^) (Table 1), but within a range of values considered adequate for substrates (0.19 to 0.75 g cm^−3^) [15,25].

Mass wetness and moisture content average values of the coir were higher than those of the mixes. In the mixes, mass wetness ranged from 4.94 to 5.35 g water/g substrate and the moisture content average ranged from 79.6% to 81.7%, *w/w*. Total porosity was higher than 85% *v/v* in all substrates, which is the value recommended for substrates [15]. Therefore, the physical characteristics of the mixes are close to or within the range of values adequate for the substrates for plant production.

In coir, perlite, pine bark, biochar, and MSW, nitrate (NO_3_-N) levels determined in aqueous extracts (1:5 substrate:water, v:v), using an ion-specific electrode and meter (Crison Instruments, Barcelona, Spain), were 0, 0.5, 12.1, 4.1, and 91 mg NO_3_ L^−1^, respectively.

### 2.2. Drainage Water

The hydronium concentration in the drainage water was affected by the substrate (Figure 1). In general, the pH was greater in the drainage water collected from the mixes than from the coir. Therefore, there was leaching of hydroxide (OH^−^), which may contribute to lowering the growing medium pH (Table 1). The great leaching of hydroxide over time took place in the mix of coir + MSW + perlite, probably due to the lower CTC of perlite. After the second sampling date, the leachate pH in general decreased over time (Figure 1).

The EC and the concentration of NO_3_ in the drainage water were also affected by the substrate (Figure 1). EC was initially greater in the mix of coir + MSW + perlite than with the other substrates (Figure 1, or in nutrient solution (1.6 ± d*S*·m^−1^)) (Table 1). Therefore, there was leaching of salts contained in the mix. This could be related to perlite since when it was changed by pine bark, the ion leaching was lower. Pine bark in the mix may reduce ion leaching due to its high cation exchange capacity (CEC) [16]. On the last three sampling dates, leachate EC was greater in the mix of coir + biochar + pine bark. The EC of leachate was lower in the mix of coir + biochar + pine bark than in the other substrates, which is consistent with the initial EC of this mix (1.5 d*S* m^−1^) (Table 1).

Overall, nitrate content was higher in drainage water from the mixes with MSW compost than from the other three substrates (Figure 1). This may be due to the higher content of nitrates of MSW or the immobilization of nitrogen in the other blends. Like the EC, the higher nitrate values occurred in the mix with perlite. In the other growing media, the leachate nitrate values during the growing period were close to the values of the nutrient solution. This indicates that the nitrogen-nitrate content in a nutrient solution for mixes containing MSW must be reduced.

### 2.3. Shoot Nutrient Concentration and Uptake

Shoot N, K, Ca, and Mn contents were significantly affected by the substrate (Table 2). Shoot N content was lower in plants grown in the mix of coir + MSW + perlite (4.68%) than those grown in the other substrates. Plants grown in the mixes with biochar had a lower shoot K concentration than plants grown in the other substrates. Shoot Ca and Mn concentrations in the plants grown in the mixes were higher than or similar to those plants grown in coir.

Shoot average values of Ca content were significantly higher in plants grown in mixes with MSW. Plants grown in mixes with MSW had a higher average Ca content in their shoots. This could be linked to the calcium concentration in the MSW (7.6% CaO). The use of perlite instead of pine bark increased shoot Ca. This may be due to an increase of macro-porosity caused by perlite [27,28], which may contribute to Ca mass flow. The shoot levels of Ca in growing media without MSW, even in coco, were low. However, in a similar study using coir as a substrate and a similar nutrient solution, but with emitters placed between crop rows instead of emitters placed at the base of each plant leaf, Ca content was within a sufficient range [2]. Therefore, the emitter position may affect Ca uptake since it affects wetting and salt patterns in root medium. Thus, further research to clarify the Ca concentration in the substrate and in drainage water must be evaluated. Despite the differences in shoot Ca content, none of the plants grown in the different mixes showed visual symptoms of Ca deficiency (tip burn of inner, newly developing leaves).

Mn levels in shoots were greater in the biochar-containing mixes than in the other mixes (Table 2). This may be due to the level of Mn in biochar. The level of Mn in biochar is influenced by particle size, however it can be high [29]. The authors of [30] reported that the blend of (*Pinus sylvestris* L.) wood biochar with nitrogen, phosphorus, and potassium (NPK) fertilizers increased the content of Mn and Fe in plants but decreased the contents of Ca and Mg.

The uptake of macronutrients by shoots (except N) was influenced by the substrate (Table 3). Relatively to plants grown in coir, the addition of perlite to biochar or to MSW increased shoot P, K, Ca, and Mg uptake. However, shoot Ca uptake in plants grown in the mix of coir + municipal waste compost + perlite was higher than those grown in the other substrates. In comparison to the other substrates, adding pine bark to the mix with biochar reduced shoot P and Mg uptake.

The Fe and Mn concentrations were affected by mixes, but their accumulation was also higher or similar to that of those plants grown in coir. Shoot Mn uptake was higher in plants grown in mixes with biochar than those grown in the other mixes (Table 3).

The results indicate that from the standpoint of mineral nutrients, the mixes (except coir + biochar + pine bark) overall allowed a nutrient accumulation similar to or high than those plants grown in coir.

### 2.4. Photosynthetic Pigments

Photosynthetic pigments were affected by the substrate (Table 4). Total chlorophyll (Chl a + Chl b) and Chl a were higher in plants grown in coir than in those grown in the mixes. Leaf total chlorophyll ranged from 24.33 to 37.36 mg 100 g^−1^ FW (fresh weight). These values were lower than previously reported ranges. This can be due to the genotype and/or growing conditions, as reported in [31].

Leaf total chlorophyll and Chl a were related to leaf Ca according to polynomial regressions: [Leaf total chlorophyll (mg.100 g^−1^ FW) = 119.66 (Leaf Ca%)^2^ − 201.41 (Leaf Ca%) + 109.62, R^2^ = 0.897 *p* < 0.001; Leaf Chl a (mg.100g^−1^ FW) = 93.558 (Leaf Ca%)^2^ − 162.59 (Leaf Ca%) + 81.66, R^2^ = 0.9703 *p* < 0.001].

Chlorophyll b content was higher in the mix of coir + MSW + perlite compared to those plants grown in the other substrates. Leaf Cc average values were higher in plants grown in coir + MSW + perlite and coir than in those grown in the other two mixes.

Leaf Cc content was also related to leaf Ca content according to polynomial regression [leaf Cc (mg.100g^−1^ FW) = 160.49 (Leaf Ca %)^2^ − 253.07(leaf Ca %) + 141.2, R^2^ = 0.90 *p* < 0.001]. Leaf chlorophyll and carotenoid content also increased with leaf Ca^2+^ excess and deficiency in *Mentha Pulegium* L. [32]. However, in *Eruca sativa* Miller, Ca deficiency led to lessening the content of the chlorophylls a + b and carotenoids (Silva et al., 2021).

Leaf Cc average values ranged from 39.69 to 54.76 mg 100 g^−1^ FW, and this range of values is higher than those reported in [33] (17–32 mg 100 g^−1^ FW) and [34] (21.5–31.1 mg 100 g^−1^ FW). This may be owing to genotype and/or growing conditions [31]. In spinach, carotenoid content is affected by water [35], nutrient uptake [36], etc. The Chl a/Chl b ratio was lower in plants grown in mixes than those grown in coir. The Chl a/Chl b ratio correlated to the leaf calcium content [(Chl a/Chl b ratio) = 4.738 (Leaf Ca %)^2^ − 8.498 (Leaf Ca %) + 4.63, R^2^ = 0.98 *p* < 0.01)].

### 2.5. Plant Growth

Shoot dry weight and fresh yield were affected by the substrate (Table 1). Plants grown in the different blends had a greater shoot dry weight (g/plant) and percentage, significantly higher than those plants grown in coir (Table 5). Shoot dry weight accumulation was higher in plants grown in the mix with MSW and perlite but was not significantly higher than in those plants grown with biochar + perlite or MSW + pine bark. Shoot dry weight (mg/plant) increased linearly with leaf calcium content according to the following equation: shoot dry weight (mg/plant) = 1.309 leaf Ca (%) + 1.71 (R^2^ = 0.94, *p* < 0.001).

Despite the high initial salinity of the compost-containing mixtures (2.8 and 3.0 d*S*·m^−1^) (Table 1), higher than the salinity threshold of spinach (2 d*S* m^−1^), plants grown in those mixes had a greater shoot dry and fresh yield than those grown in the other substrates (Table 5). In the mix of coir + MSW compost + perlite, this may be due to the decrease of salinity of the substrate owing to the initial leaching of ions (Figure 1).

However, for the substrate coir + MSW compost + pine bark, this did not occur, and the shoot dry weight and fresh yield (2.8 kg/m^2^) were also high. This can be owing to the constant high-water potential around the plant roots and/or the cultivation period that did not subject plants to high transpiration or temperature. Under these conditions, greenhouse plants can grow with higher electrical conductivities (3 to 6 d*S* m^−1^) than in the current study [37]. The authors of [38] also reported that spinach grown between January (seeding) and March can tolerate high salinity two-fold over its previously considered salinity threshold. Moreover, the influence of EC values on plant growth is also dependent on the spatial distribution of ions in the substrate. Under drip irrigation, inside the wet bulb, EC values are lower, creating a suitable root-zone salinity [39].

The leaf area of the plants grown in the biochar-containing mixtures (Table 5) was smaller than that of the plants grown on the other substrates. The leaf area decreased linearly with manganese uptake by plants [leaf area (cm^2^) = 0.3119 (µg Mn/plant) + 574.72, R = 0.987 *p* < 0.001], which was higher in plants grown in mixes with biochar (Table 3). A decrease in leaf area with the increase in Mn in the nutrient solution was reported in [40]. 

It should be noted that despite the irrigation scheduling and nutrient solution to be optimized to coir, fresh yield, in all mixes, was higher than or similar to that obtained in the coir. This indicates that the use of MSW and biochar have the potential to reduce the use of coir and peat.

### 2.6. Phytochemical Accumulation and Antioxidant Activity

#### 2.6.1. Total Phenols and Antioxidant Activities

Leaf total phenols and DPPH antioxidant activity were unaffected by the substrates (Table 6). Average total phenol level values ranged from 102.4 to 131.6 milligrams of gallic acid equivalent per 100 g of fresh weight (mg GAE/100 g FW). These values were within the range of published values for spinach (72 to 320 mg GAE/100 g FW) of different cultivars under different cultural practices, climate conditions, maturity, etc. [41,42,43].

Leaf DPPH average values ranged from 53.6 to 67.0 mg GAE/100 g FW (Table 6). These values were lower than those reported in [44] (100 mg/100 g FW) and within the range reported in [2,43,45,46].

Leaf FRAP antioxidant activity was higher in plants grown in coir (235.8 mg/Trolox/100 g FW) and coir + biochar + pine bark (211.9 Trolox mg/100 g FW) than in those grown in the other mixes (Table 6). Leaf FRAP content was also related to leaf calcium content by the equation: leaf FRAP (mg Trolox/100 g FW) = 649.91 leaf Ca (%)^2^ − 1089.1 leaf Ca (%) + 627.01, R^2^ = 0.8325 *p* < 0.01). In lettuce, the FRAP content response to different levels of calcium nitrate was also not linear [47], likely due to the Ca contained in the fertilizer.

#### 2.6.2. Ascorbate-Glutathione Cycle

The ascorbate-glutathione (AsA-GSH) pathway comprises of AsA (ascorbate), GSH (glutathione), and four enzymes that play a vital role in detoxifying ROS [48]. AsA levels conferred plants better tolerance to abiotic stresses by reducing the ROS [49] by detoxifying H_2_O_2_ via ascorbate peroxidase (APx) [50]. 

Plants grown in coir + biochar + perlite, coir + MSW + perlite, and coir + biochar + pine bark had higher leaf AsA than those grown in coir and coir + MSW + pine bark (Table 7). This suggests that these mixtures may have been exposed to more abiotic stress, which may have contributed to the synthesis and accumulation of bioactive metabolites such as ascorbate [51], allowing the plant to eliminate reactive oxygen species (ROS) such as superoxide radical and hydrogen peroxide [50].

The average AsA levels in leaves ranged from 9.06 to 16 mg/100 g FW. These values were similar to those published in [2] (11 to 40 mg/100 g FW) and Machado et al. (2020) (8 to 9 mg/100 g FW) for spinach produced under similar cultural practices.

Plants grown in coir had higher levels of APx (ascorbate peroxidase) and GSH than those grown in mixtures.

Ascorbate peroxidase, which predominantly scavenges H_2_O_2_ in the cytosol and chloroplast [50], increased under foliar macronutrient deficiency [52]. Ascorbate peroxidase was stimulated by Ca deficiency in maize [52], beans [53], and citrus seedlings [54]. In the current study, the highest leaf APx and GSH levels were also found in the plants with lower leaf Ca concentrations (0.51%) (Table 7).

As reported in [55], there are very few studies on the influence of Ca on oxidative stress and antioxidant response. Therefore, further research on this subject is required.

APx activity generally increases along with other antioxidants, such as GSH in plant tissues, and enhances plant stress tolerance [56,57,58]. APx was positively correlated with leaf GSH level (R = 0.655, *p* < 0.01).

Leaf glutathione peroxidase (GPx) was higher in plants grown in coir + biochar + pine bark than in those grown in the other substrates.

Leaf glutathione reductase (GR) activity was higher in plants grown in mixes with biochar than in those grown in the other substrates. However, perlite addition to the mix with biochar led to a higher increase in leaf GR. The increase in the GR activity was positively correlated with shoot Mn uptake (R = 0.783, *p* < 0.01). This was also reported in [59].

The mixes influenced the AsA and GSH content and the level of antioxidant enzymes of the AsA-GSH pathway. AsA and GPx values were higher than or similar to those obtained in coir.

This indicates that plants grown in mixtures had similar or better tolerance to abiotic stress modulated by the ascorbate-glutathione axis.

The findings show that leaf Ca and Mn levels have an impact on the AsA-GSH cycle.

However, more research is needed to evaluate the influence of the nutrient solution (concentration, composition, and pH), emitter position in relation to plants, and irrigation scheduling, adjusted to each mix on the Ca and Mn content in the leaves.

Despite the differences, this study revealed that using municipal compost and biochar, two renewable wastes produced locally, allows obtaining high yields and high-quality spinach while using less coir and peat.

## 3. Conclusions

Municipal compost and biochar are viable alternatives to reducing the use of coir and peat, contributing to increasing the sustainability of soilless culture in substrate. Spinach fresh yield in all blends with 12% (*v/v*) of selectively collected municipal organic compost or biochar and 10% (*v/v*) perlite or pine bark was equal to or higher than those obtained in coir. Blends with municipal compost, perlite, or pine bark increased spinach fresh yield by 28% and 13%, respectively, when compared to coir. The mixes did not affect total phenols content or DPPH antioxidant activity. Phytochemical accumulation was significantly influenced by leaf calcium content. However, as there are few studies on the effect of leaf calcium on antioxidant response, more research is needed.

## 4. Material and Methods

### 4.1. Growth Conditions and Substrates

The experiment was conducted in a greenhouse located at the “Herdade Experimental da Mitra” (38°31052″ N; 8°01005″ W), University of Évora, Portugal. The greenhouse was covered with polycarbonate and had no supplemental lighting or heating. Diurnal changes in air temperature inside the greenhouse at the plant canopy level ranged from 8 to 27 °C. Solar radiation ranged from 34 to 248 W·m^−2^·d ^−1^.

The experiment included five substrates: coir (control) and four mixtures. The mixtures were coir-based (Projar S.A. Spain) and mixed with other components. The coir (100% coir pith) had a pH of 5.5–6.0, electrical conductivity (EC) > 1.5 d*S* m^−1^, granulometry = 0–10 mm, total porosity = 95%, air (%, *v/v*) = 25, and CEC (meq/100 g) = 60–120.

Two renewable resources produced in Portugal were used as mix components: municipal solid waste compost (MSW) (Nutrimais, Lipor company, Portugal) and Acacia wood biochar (Ibero Massa, Oliveira de Azeméis, Portugal). The raw materials used in the “Nutrimais” manufacturing process include horticultural products, food scraps carefully selected from restaurants, canteens, and similar establishments, forest exploitation residues (e.g., branches and foliage), and green residues (e.g., flowers, grasses, prunings).

To improve and adjust the characteristics of the mixes, pine bark (Siro, Mira, Portugal), also a renewable product produced in Portugal, and Perlite PERLIGAN^®^ Extra (Knauf, Dortmund, Germany) were used. According to the manufacturer, pine bark is made up of 100% maritime pine bark, selected, screened, and calibrated. After sieving, it was subjected to heat treatment to eliminate the pine wood nematode (*Bursaphelenchus xylophilus*).

The MSW was produced from organic material selectively collected and the maximum heavy metal concentration was below the maximal levels permitted in different European countries [60].

MSW compost in the powdery form had (expressed as a percentage of compost dry weight): organic matter (53.2%), humidity (23.7%), humic acids (13%), C/N ratio (12.4), C (29.15%), N (2.3%), P_2_O_5_ (1.2%), K_2_O (1.81%), CaO (7.06%), MgO (0.2%), and 0.3, 28.7, 15, 51.0, 0.1, 7.6, 150.0, and 32.3 mg/kg of Cd, Pb, Cr, Cu, Hg, Ni, Zn, and B, respectively. The EC and pH (1:5 compost:distilled water, *v/v*) were 5.4 d*S*.m^−1^ and 9.0, respectively.

Acacia wood biochar from pyrolysis had a pH of 8 to 10, an EC of 0.25 d*S* m^−1^, and a granulometry of 1 to 20 mm. The pine bark had a particle size of between 8 and 15 mm, and the pH of CaCl2 was 4.5. Pine bark had (expressed as a percentage of compost dry weight): organic matter (99.1%), C/N ratio (278), C (55.6%), N (0.20%), P2O5 (0.04%), K2O (0.11%), and Mg (0.05%). The perlite was pH-neutral, chemically inert, salt-free, and had a grain size of 2 to 6 mm. 

The percentage of the different components (%, *v/v*) in the mixtures is presented in Table 8.

The physicochemical characteristics of the mixtures measured were pH, EC, mass wetness, moisture content, total porosity, and bulk density. The pH and the EC were measured in the aqueous extract (1:5 substrate:water, v:v). The aqueous extract of each mix was obtained by combining one part (by volume) of substrate with five parts (by volume) of distilled water. The sample was mixed and left to stand for 30 min to equilibrate. The mixture was poured into a clean funnel lined with a filter to avoid getting substrate in the solution. The pH and EC were measured using a pH meter (Fiveeasy, Mettler Toledo) and a conductivity meter (LF 330 WTW, Weilhein, Germany), respectively.

Physical properties of mass wetness, moisture content, total porosity, and bulk density were determined following the methodology described in [26].

Spinach (*Spinacia oleracea* L. cv. Tragopan) seedlings were produced in soil blocks, with four seedlings per block. Soil blocks were obtained from a commercial nursery. The seedlings were grown in a growing medium (90% black peat and 10% brown peat) and fertilized with 0.5 g of 8-17-27 per liter of the substrate. They were transplanted into Styrofoam plant boxes, on 23 February 2021, at 22 days after emergence. The boxes (100 × 25 × 10 cm) were filled with 14 L of each mixture at the height of approximately 7 cm. The blocks were spaced at 12.5 cm in two rows per box and 10 cm between rows with a plant density of 256 plants m^−2^.

Treatments were arranged in a complete randomized block design with five replicates. Each planting box was irrigated using 8 L·h ^−1^ pressure compensating and anti-drain emitters. The emitters were attached to 4 fine tubes of 70 cm in length and 5 mm in diameter, inserted into the substrate near the plant base. Thus, 8 water emission points were used per box.

The irrigation schedule was optimized for coir. It was based on substrate volumetric water content at the Styrofoam box control (coir) measured using a soil moisture probe (SM105T delta devices England), and the volume of water was drained. The nutrient solution was applied 3 to 7 times per day and averaged 10% to 35% of drainage (leaching fraction) for each application. The leaching fraction was controlled through a relay level connected to an electric valve that stopped watering when the level of leached water was within 10% to 25% of the applied water. The nutrient solution, except in the first irrigation, to moisten the growing medium was applied in each irrigation from transplanting to the day before harvesting.

Initially, the solution contained 8.34 mmol L^−1^ NO_3_-N, 2.71 mmol L^−1^ NH_4_-N, 0.68 mmol L^−1^ P, 3.88 mmol L^−1^ K, 2.89 mmol L^−1^ Ca, 1.49 mmol L^−1^ Mg, 0.80 mmol L^−1^ S, 46 µmol L^−1^ B, 7.86 µmol L^−1^ Cu chelated by EDTA, 8.95 µmol L^−1^ Fe chelated by EDTA, 18.3 µmol L^−1^ Mn chelated by EDTA, 1 µmol L^−1^ Mo, 2 µmol L^−1^ Zn chelated by EDTA, 2.1 mmol L^−1^ Cl, and 0.7 mmol L^−1^ Na. The EC value of the nutrient solution ranged over time was: 1.2 ± 0.2 d*S* m^−1^ (from transplanting to 6 days after transplanting, DAT), 1.4 ± 0.2 d*S* m^−1^ (from 7 to 20 DAT), and 1.8 ± 0.2 d*S* m^−1^ (from 21 to 30 DAT). The EC values were obtained by changing the injection rate of the nutrient solution. 

At 31 DAT, to reduce the nitrate concentration in the leaves, the nutrient concentration and the NO_3_/NH_4_ ratio in the nutrient solution were adjusted to 4.26 mmol·L^−1^ NO_3_-N, 4.11 mmol·L^−1^ NH_4_-N, 0.67 mmol·L-1 P, 2.84 mmol·L^−1^ K, 2.13 mmol·L^−1^ Ca, 0.88 mmol·L^−1^ Mg, 0.47 mmol·L^−1^ S, 46 µmol L^−1^ B, 7.86 µmol L^−1^ Cu chelated by EDTA, 8.95 µmol L^−1^ Fe chelated by EDTA, 18.3 µmol L^−1^ Mn chelated by EDTA, 1 µmol L^−1^ Mo, 2 µmol L^−1^ Zn chelated by EDTA, 2.1 mmol L^−1^ Cl, and 0.7 mmol L^−1^ Na. The EC of the nutrient solution was 1.9 ± 0.2 d*S* m^−1^.

### 4.2. Measurements

The pH, EC, and the concentration of NO_3_^-^ of the drainage water from each box were measured weekly using a potentiometer (pH Micro 2000 Crison), a conductivity meter (LF 330 WTW, Weilheim, Germany), and an ion-specific electrode (Crison Instruments, Barcelona, Spain), respectively, following the procedures outlined in [61]. 

The plants were harvested at 34 DAT (28 March 2021). The shoots of the plants were cut off at 1 cm above the substrate surface and rinsed with distilled water. Five sample plants (shoots) from each box were washed, oven-dried at 70 °C for 2–3 days, weighed, ground so that they would pass through a 40-mesh sieve, then analyzed for N, P, K, Ca, Mg, Na, B, Cu, Mn, and Zn. Total N was analyzed by using a combustion analyzer (Leco Corp. St. Josef, MI, USA). The K and Na were analyzed by flame photometry (Jenway, Dunmow, UK). The P and B were analyzed using a UV/Vis spectrometer (Perkin Elmer lamba25, USA). The remaining nutrients were analyzed using an atomic absorption spectrometer (Perkin Elmer, Inc., Shelton, CT, USA).

Samples of 1000 g of spinach leaf-blade from each box were macerated in a mortar and homogenized in 8 mL of methanol/water solution (90:10 (*v/v*), MW90 extract) for 1 min and then centrifuged at 4 °C at 6440× *g* for 5 min [62]. The extracts were stored in aliquots at −20 °C for later use.

Total chlorophyll, chlorophyll a (Chl a) and b (Chl b), and total carotenoids (Cc) were determined in MW90 extract, as previously described in [2].

Samples of 1000 g of spinach leaf-blade were macerated in a mortar and homogenized in 8 mL of methanol/water solution (80:20 (*v/v*), MW80 extract) for 1 min and then centrifuged at 4 °C at 6440× *g* for 5 min. The extracts were stored in aliquots at −20 °C for later use.

The content of total phenolic compounds (TPC), ascorbate (AsA), as well as the 2,2-diphenyl-1-picrylhydrazyl free radical scavenging antioxidant power (DPPH) and ferric-reducing antioxidant power (FRAP) of MW80 extract were determined using the methodology described in [2].

Samples of 1000 g of spinach leaf-blade were macerated in liquid N_2_ and homogenized in 5 mL of 0.12 mM phosphate buffer, pH 7.2. The supernatant obtained after centrifugation of this extract for 15 min at 15,000× *g* at 4 °C was collected and stored in aliquots at −20 °C (PB extract) [63].

Glutathione (GSH), protein content, glutathione reductase (GR), and peroxidase (POx) enzyme activities were determined in PB extract in accordance with the methodology described in [2]. 

Glutathione peroxidase (GPx) enzyme activity was determined in PB extract in accordance with [64] in a reaction mixture containing 5 mM of GSH, 0.24 U/mL of GR, 0.16 mM of NADPH, and a suitable volume of leaf-blade PB extract in 0.12 mM of phosphate buffer, pH 7.2. The reaction mixture was incubated for 5 min at 37 °C and the oxidation of NADPH was determined by reading the absorbance at 340 nm for 180 s at 37 °C. The reaction was then started with the addition of t-butyl hydroperoxide (t-BHP) and was followed by reading the absorbance at 340 nm for 180 s at 37 °C. GPx enzyme activity was calculated based on the slope of the reaction curves, using an extinction coefficient value of 6.22 mM^−1^ cm^−1^ for NADPH. GPx activity was expressed in terms of nmol min^−1^/mg protein.

Ascorbate peroxidase (APx) activity was determined in PB extract in accordance with the Janda method [65], which is based on the breakdown of ascorbate coupled with the reduction of H_2_O_2_ to H_2_O. The reaction was started by mixing 1 mM of ascorbate, 10 mM of H_2_O_2_, 0.1 mM of EDTA, and a suitable volume of leaf-blade PB extract in 50 mM of phosphate buffer, pH 7.0. The ascorbate consumption was followed by reading the absorbance at 290 nm, for 180 s at 37 °C. APx enzyme activity was calculated based on the slope of the reaction curves, using an extinction coefficient value of 2.8 mM^−1^.cm^−1^ for ascorbate.

### 4.3. Data Analysis

Data were analyzed by analysis of variance using SPSS Statistics 25 software (Chicago, IL, USA). Means were separated at the 5% level using Duncan’s new multiple range test. Bivariate correlation analysis between parameters was realized using Pearson’s bilateral correlation coefficient. 

## Figures and Tables

**Figure 1 plants-11-01893-f001:**
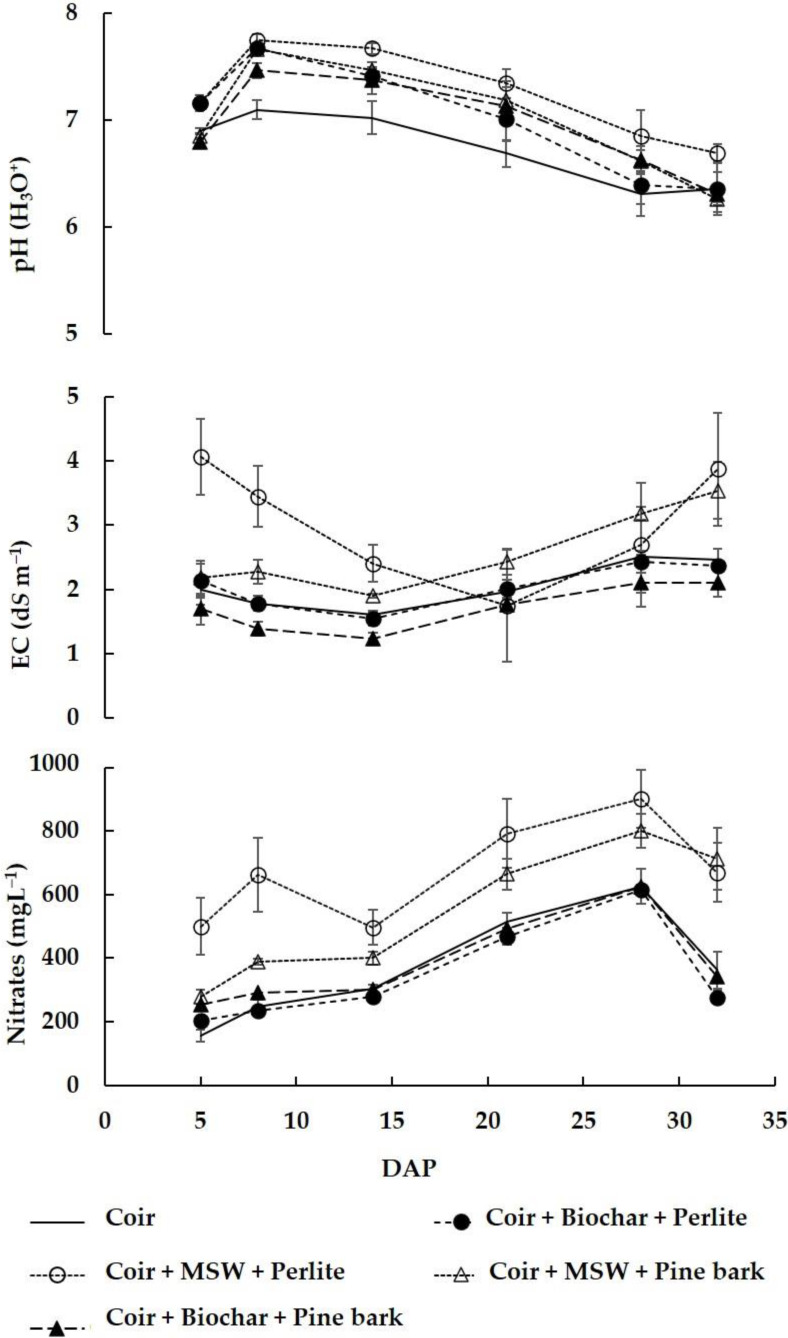
pH, EC, and NO_3_^-^ concentrations in the drainage water. Each symbol represents the mean of five replicates, and the error bars represent ± 1SE.

**Table 1 plants-11-01893-t001:** Initial physicochemical properties of the mixes.

Substrate	Coir	Coir + Biochar + Perlite	Coir + MWS + Perlite	Coir + Biochar + Pine Bark	Coir + MSW + Pine Bark
pH *	5.65 ^c†^	6.52 ^b^	7.10 ^a^	6.77 ^b^	7.00 ^a^
EC (d*S*·m^−1^) *	1.8 ^b^	1.51 ^c^	2.80 ^a^	1.12 ^c^	3.10 ^a^
Total porosity (*v/v*. %) **	90.2 ^c^	97.0 ^a^	98.2 ^a^	95.7 ^b^	96.3 ^b^
Moisture content (*w/w*, %) **	95.0 ^a^	81.1 ^b^	79.6 ^b^	78.9 ^b^	81.7 ^b^
Mass wetness (g water/g substrate) **	7.90 ^a^	5.39 ^b^	5.01 ^b^	5.00 ^b^	4.94 ^b^
Bulk density (g·cm^−^^3^) **	0.11 ^b^	0.17 ^a^	0.18 ^a^	0.17 ^a^	0.19 ^a^

* Determined in the aqueous extract (1:5 substrate:water, v:v). ** Determined following the methods described in [26]. Moisture content: The percent of moisture found in a sample on a wet mass basis. This is calculated by ((wet weight − dry weight)/wet weight) × 100. Mass wetness: The water content of a sample on a dry mass basis. This is calculated by (wet weight − dry weight)/dry weight. ^†^ Means followed by different letters within a line are significantly different at *p* < 0.05.

**Table 2 plants-11-01893-t002:** Effects of substrates on shoot nutrient concentration.

Substrate	Shoot Macronutrients (%)					Shoot Micronutrients (μg·g^−1^)				
	N	P	K	Ca	Mg	Fe	B	Mn	Zn	Na ^1^
Coir	5.87 ^a†^	0.56 ^a^	6.01 ^a^	0.51 ^d^	0.35 ^a^	141.3 ^a^	21.6 ^a^	85.6 ^b^	56.3 ^a^	0.84 ^a^
Coir + biochar + perlite	5.69 ^a^	0.54 ^a^	5.37 ^b^	0.72 ^bc^	0.35 ^a^	147.5 ^a^	26.4 ^a^	166.9 ^a^	60.0 ^a^	0.66 ^a^
Coir + MWS + perlite	4.68 ^b^	0.59 ^a^	5.67 ^ab^	1.07 ^a^	0.30 ^a^	160.6 ^a^	18.7 ^a^	61.87 ^c^	58.0 ^a^	0.91 ^a^
Coir + biochar + pine bark	5.75 ^a^	0.51 ^a^	5.28 ^b^	0.64 ^bc^	0.26 ^a^	138.7 ^a^	20.5 ^a^	148.1 ^a^	75.6 ^a^	0.80 ^a^
Coir + MWS + pine bark	5.40 ^a^	0.61 ^a^	5.70 ^ab^	0.84 ^b^	0.26 ^a^	133.8 ^a^	17.2 ^a^	60.6 ^c^	61.9 ^a^	0.88 ^a^
Significance	*	NS	*	***	NS	NS	NS	***	NS	NS

^†^ Means followed by different letters within a column are significantly different at *p* ≤ 0.05. NS: nonsignificant, *, and ***: significant at *p* < 0.05, 0.01, and 0.001 levels, respectively. ^1^ Although sodium is not a micronutrient, it is included here for convenience.

**Table 3 plants-11-01893-t003:** Effects of the substrates on shoot nutrients’ uptake.

Substrate	Shoot Macronutrients’ Uptake (mg/Plant)					Shoot Micronutrients’ Uptake (µg/Plant)				
	N	P	K	Ca	Mg	Fe	B	Mn	Zn	Na ^1^
Coir	135.0 ^a^^†^	2.91 ^b^	137.1 ^b^	11.5 ^c^	2.02 ^b^	326.6 ^b^	49.9 ^a^	196.4 ^c^	129.4 ^a^	1.9 ^a^
Coir + biochar + perlite	155.2 ^a^	4.15 ^ab^	146.9 ^ab^	19.8 ^b^	2.50 ^a^	400.9 ^ab^	71.2 ^a^	455.3 ^a^	163.9 ^a^	1.8 ^a^
Coir + MWS + perlite	143.8 ^a^	4.53 ^a^	173.5 ^a^	32.9 ^a^	2.88 ^a^	495.2 ^a^	57.4 ^a^	191.0 ^c^	179.8 ^a^	2.8 ^a^
Coir + biochar + pine bark	146.8 ^a^	2.22 ^c^	135.5 ^b^	16.4 ^bc^	1.70 ^c^	357.1 ^ab^	52.4 ^a^	379.6 ^b^	192.9 ^a^	2.0 ^a^
Coir + MWS + pine bark	151.3 ^a^	3.13 ^b^	159.5 ^ab^	23.7 ^b^	2.01 ^b^	377.2 ^ab^	48.2 ^a^	169.7 ^c^	171.8 ^a^	2.5 ^a^
Significance	NS	***	*	***	**	*	NS	***	NS	NS

^†^ Means followed by different letters within a column are significantly different at *p* ≤ 0.05. NS: nonsignificant, *, **, and ***: significant at *p* < 0.05, 0.01, and 0.001 levels, respectively. ^1^ Although sodium is not a micronutrient, it is included here for convenience.

**Table 4 plants-11-01893-t004:** Effect of substrates on leaf photosynthetic pigments’ content and Chl a/Chl b ratio.

Substrate	Photosynthetic Pigments (mg/100 g FW)				
	Total Chl	Chl a	Chl b	Cc	Chl a/Chl b Ratio
Coir	37.36 ^a†^	22.55 ^a^	14.81 ^ab^	54.59 ^ab^	1.51 ^a^
Coir + biochar + perlite	25.34 ^c^	12.70 ^bc^	12.63 ^b^	42.07 ^bc^	1.01 ^b^
Coir + MSW + perlite	31.44 ^b^	14.99 ^bc^	16.45 ^a^	54.76 ^a^	0.91 ^b^
Coir + biochar + pine bark	32.75 ^b^	17.24 ^ab^	15.51 ^ab^	48.07 ^abc^	1.14 ^b^
Coir + MSW + pine bark	24.03 ^c^	10.51 ^c^	13.52 ^b^	39.69 ^c^	0.78 ^b^
Significance	**	***	*	*	***

^†^ Means followed by different letters within a column are significantly different at *p* ≤ 0.05. *, **, and ***: significant at *p* < 0.05, 0.01, and 0.001 levels, respectively.

**Table 5 plants-11-01893-t005:** Effects of substrates on shoot dry weight, leaf area, and fresh yield of spinach.

Substrate	Shoot Dry Weight		Leaf Area	Fresh Yield
	(g/Plant)	(%)	(cm^2^/Plant)	(kg/m^2^)
Coir	2.30 ^c†^	8.88 ^b^	510.9 ^a^	2.52 ^b^
Coir + biochar + perlite	2.73 ^ab^	9.50 ^a^	427.7 ^b^	2.75 ^b^
Coir + MWS + perlite	3.05 ^a^	9.05 ^a^	514.3 ^a^	3.25 ^a^
Coir + biochar + pine bark	2.56 ^bc^	9.40 ^a^	463.6 ^b^	2.61 ^b^
Coir + MWS + pine bark	2.86 ^ab^	9.39 ^a^	523.0 ^a^	2.87 ^ab^
Significance	*	*	*	*

^†^ Means followed by different letters within a column are significantly different at *p* ≤ 0.05. * Significant at *p* < 0.05.

**Table 6 plants-11-01893-t006:** Effect of substrates on total phenols content and antioxidant activity.

Substrate	TPC (mg GAE/100 g FW)	DPPH (mg GAE/100 g FW)	FRAP (mg Trolox/100 g FW)
Coir	125.2 ^a^	67.0 ^a^	235.8 ^a^
Coir + biochar + perlite	122.9 ^a^	57.6 ^a^	161.2 ^c^
Coir + MSW + perlite	131.6 ^a^	53.6 ^a^	202.4 ^b^
Coir + biochar + pine bark	130.3 ^a^	64.6 ^a^	211.9 ^ab^
Coir + MSW + pine bark	102.4 ^a^	55.4 ^a^	172.6 ^c^
Significance	NS	NS	***

Means followed by different letters within a column are significantly different at *p* ≤ 0.05. NS: nonsignificant, ***: significant at *p* < 0.001.

**Table 7 plants-11-01893-t007:** Effect of substrates on the ascorbate-glutathione cycle.

Substrate	APx a	GPx	GR	AsA	GSH
	(nmol min^−1^/mg Protein)			(mg/100 g FW)	
Coir	48.43 ^a^	59.6 ^b^	9.46 ^c^	10.77 ^b^	3.63 ^a^
Coir + biochar + perlite	24.79 ^b^	58.5 ^b^	17.67 ^a^	15.71 ^a^	2.76 ^b^
Coir + MSW + perlite	18.86 ^b^	73.8 ^b^	9.71 ^c^	14.41 ^a^	1.85 ^c^
Coir + biochar + pine bark	5.34 ^c^	107.6 ^a^	12.98 ^b^	14.38 ^a^	1.82 ^c^
Coir + MSW + pine bark	21.29 ^b^	68.8 ^b^	11.26 ^c^	9.06 ^b^	2.32 ^bc^
Significance	***	***	***	***	***

Means followed by different letters within a column are significantly different at *p* ≤ 0.05. *** Significant at *p* < 0.001.

**Table 8 plants-11-01893-t008:** Substrate components (%, *v/v*).

Substrate	Substrate Components (%, *v/v*)				
	Coir	MSW	Biochar	Perlite	Pine Bark
Coir	100	-	-	-	-
Coir + biochar + perlite	78	-	12	10	-
Coir + MSW + perlite	78	12	-	10	-
Coir + biochar + pine bark	78	-	12	-	10
Coir + MSW + pine bark	78	12	-	-	10

## Data Availability

Data is contained within the article.

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
