# Peer review of "Effects of Coir-Based Growing Medium with Municipal Solid Waste Compost or Biochar on Plant Growth, Mineral Nutrition, and Accumulation of Phytochemicals in Spinach"

_plants, 2022, doi:10.3390/plants11141893_

Round 1
Reviewer 1 Report
1.Please indicate the nutrient content (N, P, K, Ca, Mg, etc., part of availability) of the five substrates (coir, biochar, perlite, MSW, pine bark) and whether the nutrients in these subtrates were possible during the test is released, or the added nutrient solution is adsorbed.
2.It can be seen from the table that there are few significant differences in the results of many trials. What are the important findings of the authors? What important ideas and knowledge can readers get from the results?
3.There are many related studies at present. What are the author's outstanding insights and findings? How to distinguish it from previous studies?
Author Response
Dear Reviewer,
We want to thank you for the care and attention that you showed in the evaluation of our article entitled “Effects of coir-based growing media with municipal solid waste compost or biochar on plant growth, mineral nutrition, and phytochemical accumulation in spinach”. We have extensively analyzed each of your valuable suggestions and enlightening criticisms, which motivated us to make a substantial improvement in the article initially submitted to Plants. In this letter, we will explain how we followed each one and all your priceless suggestions for improvement. It is, therefore, our perception that the version we now resubmit for your new analysis reveals strong improvements.
- Please indicate the nutrient content (N, P, K, Ca, Mg, etc., part of availability) of the five substrates (coir, biochar, perlite, MSW, pine bark) and whether the nutrients in these substrates were possible during the test is released, or the added nutrient solution is adsorbed.
Response: We understand your question, but we have not determined nutrient availability. We only determined the nitrate content in the substrates before the experiment. “In coir, perlite, pine bark, biochar, and MSW, nitrate (NO3-N) levels determined in aqueous extracts (1:5 substrate: water, v:v) using an ion-specific electrode and meter (Crison Instruments, Barcelona, Spain), were 0, 0.5, 12.1, 4.1, and 91 mg NO3 L-1, respectively”
To address the question "possible during the test is released, or the added nutrient solution is adsorbed." We added a sentence to the text: " This may be due to higher content of nitrates of MSW or immobilization of nitrogen in the other blends.”
We also introduce a sentence with pine bark characteristics “Pine bark had (expressed as a percentage of compost dry weight): organic matter (99.1 %), C/N ratio (278), C (55.6%), N (0.20%), P2O5 (0.04%), K2O (0.11%), and Mg (0.05%).”
It can be seen from the table that there are few significant differences in the results of many trials. What are the essential findings of the authors? What important ideas and knowledge can readers get from the results?
The text has been altered in various places to attempt to respond to the query.
3. There are many related studies at present. What are the author's outstanding insights and findings? How do distinguish it from previous studies?
Response: We changed some sentences in different sections of the manuscript and we think we addressed your two previous questions
For example, the end of the discussion has been changed, “Despite the differences, this study reveals that using municipal compost and biochar, two renewable wastes locally produced, allows obtaining high yields and high-quality spinach while using less coir and peat”.
The text of the conclusions has been rewritten to highlight the findings, “Municipal compost and biochar are viable alternatives to reducing the use of coir and peat, contributing to increasing the sustainability of soilless culture in the substrate. Spinach fresh yield in all blends with 12 % (v/v) of selectively collected municipal organic compost or biochar and 10 % (v/v) perlite or pine bark was equal to or higher than those obtained in coir. Blends with municipal compost, perlite, or pine bark increased spinach fresh yield by 28 and 13 per cent, respectively, when compared to coir. The mixes did not affect total phenols content and DPPH antioxidant activity. Phytochemical accumulation was significantly influenced by leaf calcium content. However, as there are few studies on the effect of leaf calcium on antioxidant response, more research is needed.”
Reviewer 2 Report
The introduction is made for inspection and for acceptance. The results and their reports are referenced to the literature. Charts and drawings legible and understandable. Clear conclusions. Methodology correctly presented and without reservations. I am asking for the editorial correction and redaction of the text in terms of the language.
Author Response
Dear Reviewer,
We would like to thank you for the care and attention that you showed in the evaluation of our article titled "Effects of coir-based growing media with municipal solid waste compost or biochar on plant growth, mineral nutrition, and phytochemical accumulation in spinach". We have carefully considered your valuable comments, which motivated us to make a substantial improvement to the article initially submitted to Plants. Therefore, in our opinion, the version we are now resubmitting for your new analysis shows significant improvements.
- The introduction is made for inspection and for acceptance. The results and their reports are referenced to the literature. Charts and drawings legible and understandable. Clear conclusions. Methodology correctly presented and without reservations. I am asking for the editorial correction and redaction of the text in terms of the language.
Response:
Thank you very much for your comments. We made some language changes Throughout the manuscript.
Reviewer 3 Report
I am quite impressed from the experimental design and from the discussion and the conclusion. But I have some comments; I hope the authors did some.
The title: Effects of coir-based growing media with municipal solid waste compost or biochar on plant growth, mineral nutrition, and phytochemical accumulation in spinach. Should be changed into; Effects of coir-based growing medium with municipal solid waste compost or biochar on plant growth, mineral nutrition, and accumulation of phytochemical in spinach
Abstract
The aim of this study was to assess the suitability of selectively collected municipal solid waste compost (MSW) and biochar as components of the coir-based substrate to spinach grow. the suitability should changed into availability
Materials and methods
Page 10 line 360, soil blocks, the author should indicate the soil analysis of the soil box used in seedling growing.
Page 10 and page 11 lines from 395 to 445
All the physiological parameters were measured biochemically and that is good, but if the authors examined some of polyphenols genes, polyphenol oxidase, glutathione transreductase, transcription factor, PR 8 or PR 5 using real time PCR will add something good for the manuscript. It means molecular physiology is most modern way to examine the plant physiology and plant immunology.
Authors made all the measurement on the shoot system and they did not measure any parameters on the root system. Also, the length and dry weight and fresh weight etc., is very important parameters which considered as indicator on the plant growth and health.
Authors did not mention something about the major component of the waste plants they used in construction the media under the study. For example, what is the man component in coir, and or the pine park etc.,
Introduction
Authors did not mention anything about the economic value of the used medium in the study and why they selected the wastes to approach this study.
Results
Results are missed the molecular analysis of the polyphenols, some physiological enzymes using real time PCR, to examine the gene expression which could be induced in the growing medium compared with the control.
Conclusion
Needs to be rewrite especially the last two lines 310-311 should be deleted.
Author Response
Dear Reviewer,
We want to thank you for the care and attention that you showed in the evaluation of our article with title "Effects of coir-based growing media with municipal solid waste compost or biochar on plant growth, mineral nutrition, and phytochemical accumulation in spinach," as well as for your thoughtful and thorough review. We carefully considered all your valuable suggestions and enlightening criticism, which inspired us to make a significant change in the article that was initially submitted to Plants. In this letter, we will explain how we followed each one of your priceless suggestions for improvement. Therefore, in our opinion, the version we are now resubmitting for your new analysis shows significant improvements.
- The title: Effects of coir-based growing media with municipal solid waste compost or biochar on plant growth, mineral nutrition, and phytochemical accumulation in spinach. Should be changed into; Effects of coir-based growing medium with municipal solid waste compost or biochar on plant growth, mineral nutrition, and accumulation of phytochemicals in spinach
Response: We agree. The title was changed accordingly to your proposal.
2. The aim of this study was to assess the suitability of selectively collected municipal solid waste compost (MSW), and biochar as components of the coir-based substrate for spinach grown. The suitability should change into availability.
Response: If we understand the suggestion well, we prefer to maintain the term suitability since we are testing the aptness of MSW and biochar as components of coir-based substrates because we have availability of these resources.
Materials and methods
- Page 10 line 360, soil blocks, the author should indicate the soil analysis of the soil box used in seedling growing.
Response: a new sentence was added “The seedlings were grown in a growing medium (90% black peat and 10% brown peat) fertilized with 0.5 g of 8-17-27 per liter of the substrate”
- Page 10 and page 11 lines from 395 to 445
All the physiological parameters were measured biochemically and that is good, but if the authors examined some of polyphenols genes, polyphenol oxidase, glutathione transreductase, transcription factor, PR 8 or PR 5 using real time PCR will add something good for the manuscript. It means molecular physiology is most modern way to examine the plant physiology and plant immunology.
Response: In this study, we did not aim to evaluate the influence of the different mixes on the transcriptome level, but on the functional proteome level, in terms of enzyme activity and metabolome, so we did not realize qRT-PCR. In future studies, we hope to evaluate how the mixes may influence the spinach genome and the regulation of its transcription.
- Authors made all the measurement on the shoot system and they did not measure any parameters on the root system. Also, the length and dry weight and fresh weight etc., is very important parameters which considered as indicator on the plant growth and health.
Response: We are aware of the significance of the root system's length, dry weight, and fresh weight, but we did not measure these parameters, because we are going to analyze the effect of reusing these substrates on plant growth in the next experiment.
- Authors did not mention something about the major component of the waste plants they used in construction the media under the study. For example, what is the man component in coir, and or the pine park etc.
Response: It was added a sentence about the raw material used in manufacturing the municipal waste compost “The raw materials used in the “Nutrimais” manufacturing process include horticultural products; food scraps carefully selected from restaurants, canteens, and similar establishments; forest exploitation residues (e.g., branches and foliage); and green residues (e.g., flowers, grasses, prunings)”.
A sentence about perlite type and pine bark major component was added. “To improve and adjust the characteristics of the mixes, pine bark (Siro, Mira, Portugal), also a renewable product produced in Portugal, and Perlite PERLIGAN® Extra (Knauf, Dortmund, Germany), were used. According to the manufacturer, pine bark is made up of 100% maritime pine bark, selected, screened, and calibrated. After sieving, it was subjected to heat treatment to eliminate the pine wood nematode (Bursaphelenchus xylophilus)”.
Introduction
- Authors did not mention anything about the economic value of the used medium in the study and why they selected the wastes to approach this study.
Response: We agree. The sentence was rephrased and improved “A strategy to minimize or replace peat and coir could be achieved using selectively collected municipal solid waste compost (MSW) and biochar as substrate components. They are renewable resources produced locally, and the selective collection of municipal solid organic waste is increasing in Portugal. Their use lessens the carbon footprint, keeps organic waste out of landfills, and lessens Portugal's reliance on importing peat and coir”.
Results
- Results are missed the molecular analysis of the polyphenols, some physiological enzymes using real time PCR, to examine the gene expression which could be induced in the growing medium compared with the control.
Response: We did not perform qRT- PCR, because our goal was to quantify enzymes and secondary metabolites.
Conclusion
- Needs to be rewrite especially the last two lines 310-311 should be deleted.
The two lines were deleted 310 -311 were deleted
Response: The conclusions were rewritten and the sentence was deleted “Leaf ascorbate and glutathione peroxidase values in mixes were higher or similar to those obtained in coir”.
Reviewer 4 Report
1- It is required to show the importance of the study in the introduction to the abstract. and the most important recommendations must be highlighted at the end of the abstract, which justifies the practical importance of the study.
2- In line 37 “In different studies, [2] and [3] have shown that coir can”
I suggest to write in this format: “In different studies e.g., Machado, [2] and Barcelos [3] have shown that coir can”
Figure 1 The graphics are lackluster you may use black lines to improve the charts qauility.
1- The ‘discussion’ section is quite comprehensive; however, the authors should consider more about logic of the current work.
Author Response
Dear Reviewer,
We would like to thank you for taking the time to read and consider our article entitled "Effects of coir-based growing media with municipal solid waste compost or biochar on plant growth, mineral nutrition, and phytochemical accumulation in spinach," and for your thoughtful and thorough review. We carefully considered all your insightful criticisms and ideas, which inspired us to make a significant change in the article that was initially submitted to Plants. We will describe how we implemented each of your invaluable suggestions for improvement in this letter. Therefore, in our opinion, the version we are now resubmitting for your new analysis shows significant improvements.
- It is required to show the importance of the study in the introduction to the abstract. and the most important recommendations must be highlighted at the end of the abstract, which justifies the practical importance of the study.
Response: We changed the abstract
The use of municipal solid waste compost (MSW) and biochar, two renewable resources with a low carbon footprint as components of substrates, may be an alternative to reducing peat and coir usage. The aim of this study was to assess the suitability of selectively collected MSW and biochar as components of the coir-based substrate for spinach grown. An experiment was carried out to evaluate five substrates, coir and four coir-based blends (Coir + Biochar + perlite, Coir + MSW + perlite, Coir + Biochar + pine bark, and Coir + Biochar + pine bark) with 12% (v/v) MSW or biochar and 10% (v/v) perlite or pine bark. Spinach seedlings were transplanted into Styrofoam planting boxes filled with the substrate. Each planting box was irrigated daily by drip with a complete nutrient solution. Plants grown with MSW had a higher content of calcium. Shoot Mn increased in the biochar-containing mixes. The shoot dry weight of the plants grown in the different blends was higher than those grown in coir. The fresh yield was higher in mixes with MSW and perlite (3 kg/m2) or pine bark (2.87 kg/m2). Total phenols and DPPH antioxidant activity were not affected by the substrates. However, shoot AsA content was higher or equal to those plants grown in coir. MSW and biochar are alternatives to reduce the use of coir and peat.
- In line 37 “In different studies, [2] and [3] have shown that coir can”
I suggest to write in this format: “In different studies e.g., Machado, [2] and Barcelos [3] have shown that coir can”
Response: We agree. The sentence was changed in accordance with your suggestion.
- Figure 1 The graphics are lackluster you may use black lines to improve the charts qauility.
Response: The graphic quality was improved.
- The ‘discussion’ section is quite comprehensive; however, the authors should consider more about logic of the current work.
Response: We made some changes to the manuscript that we believe contribute to responding to your question.
Round 2
Reviewer 1 Report
The content of this article should be helpful to the development of related agriculture in the region, so it is recommended to submit it to a regional journal.